# Biological control under climate change: Distribution patterns of the South American fruit fly, *Anastrepha fraterculus* and two of its parasitoids in the Americas

Jesús H. Gómez-Llano[1]*, Fábio L. Galvão-Silva[1], Flor E. Acevedo[2]*, Fabio Castro-Llanos[3], Marco Silva Gottschalk[4], Dori Edson Nava[5]

1 Universidade Federal de Pelotas (UFPEL), Av. Eliseu Maciel – Campus Universitário Capão do Leão. Capão do Leão, Rio Grande do Sul, Brazil, 2 Department of Entomology, The Pennsylvania State University, University Park, Pennsylvania, United States of America, 3 International Center for Tropical Agriculture (CIAT), Valle del Cauca, Colombia, 4 Departamento de Ecologia, Zoologia e Genética, Universidade Federal de Pelotas (UFPel), Capão do Leão, Rio Grande do Sul, Brazil, 5 Embrapa Clima Temperado, Pelotas, Rio Grande do Sul, Brazil.

* jesushernandogll@gmail.com, fea5007@psu.edu

## Abstract

Climate change affects the distribution of insects, such as pests and parasitoids. Species Distribution Models (SDMs) have been developed to determine distribution patterns and risk areas for pests and biological control agents under different climate change scenarios. The South American fruit fly, *Anastrepha fraterculus* (Wiedemann) (Diptera: Tephritidae), is an important pest of cultivated fruits throughout the Americas that can be controlled by natural enemies, such as the native parasitoid *Doryctobracon areolatus* (Szépligeti) (Hymenoptera: Braconidae) and the introduced parasitoid *Diachasmimorpha longicaudata* (Ashmead) (Hymenoptera: Braconidae). However, the control efficacy and parasitism performance of these organisms could be affected by changing environmental conditions. SDMs were conducted using Random Forest to predict suitable areas for the establishment of *A. fraterculus*, *D. areolatus*, and *D. longicaudata* under different climate scenarios or Representative Concentration Pathways (SSPs) (SSP 2–4.5 and 5–8.8) in two different periods (2021–2040 and 2041–2060). Our results predicted an increase in suitable areas for *A. fraterculus* in the Americas, especially in some South American countries such as Colombia and Brazil. Moreover, the projected distribution of these species is intricately linked to the regional climatic patterns. Temperate and tropical areas were more suitable for the establishment of *A. fraterculus*; *D. areolatus* was better suited to temperate climates; while tropical climates were more suitable for *D. longicaudata*. Suitable areas for the establishment of both parasitoid species were predicted to increase in future climate scenarios, with *D. longicaudata* having a greater

**Data availability statement:** All relevant data are within the manuscript and its Supporting Information files. The raw data are publicly available in the Zenodo repository at the following DOI: https://doi.org/10.5281/zenodo.15299850.

**Funding:** The author J. H. Gomez received financial support from the Coordination for the Improvement of Higher Education Personnel (CAPES) – Financing Code 001 for the development of this study. The author D. E. Nava received financial support from the National Council for Scientific and Technological Development (CNPq), Grant #310233/2020-8, for the development of this study The funders had no role in study design, data collection and analysis, decision to publish, or preparation of the manuscript.

**Competing interests:** The authors have declared that no competing interests exist.

geographical expansion than *D. areolatus.* These parasitoids could be used as biocontrol agents in almost all areas suitable for the establishment of *A. fraterculus*.

## Introduction

Climate change and extreme weather conditions are consequences of increasing concentrations of greenhouse gases in the atmosphere from human activity [1,2]. Therefore, increases in mean temperature, fewer occurrences of freezing temperatures, and increases in extreme precipitation are expected in some areas of the planet [3]. These phenomena affect not only the weather but also the distribution of species, including insect pests and their natural enemies. Increasing temperatures can directly affect insect reproduction, development, survival, and dispersal patterns [4]. Climate change is predicted to have a greater impact on insect natural enemies, such as parasitoids, than on insect pests themselves. The limited plasticity of natural enemies at high temperatures causes temporal desynchronization with their hosts and decreases suitable distribution areas for parasitoids [5–7]. Although increasing temperatures often increase parasitism rates to a certain level, after the optimum peak is reached, any increase in temperature reduces parasitism rates [8,9]. Therefore, climate change is critical to the success of biological control programs.

The recognized effect of climate change on the distribution of insects has led to the development of computational models for predicting pest risks in different areas [10]. These species distribution models (SDMs) use environmental variables and records of species occurrences to predict potentially suitable areas for species to grow and develop; these correlative models are commonly used to predict changes in biodiversity in various climate change scenarios [11]. Currently, some SDMs are being developed using machine learning algorithms, such as MaxEnt, Bioclim, and Random Forest [12–16]. The goal of these SDMs is to predict suitable areas for species in the current and future climate scenarios. This approach may help entomologists to predict the effects of climate change on agricultural pests and their natural enemies and can be used as a tool to determine appropriate areas for the release of parasitoids for pest mitigation [17–19].

The South American fruit fly *Anastrepha fraterculus* (Wiedemann, 1830) (Diptera: Tephritidae) is one of the most significant pests in the Americas and is distributed from Mexico to Argentina [20]. Currently, this species exhibits eight distinct morphotypes, as defined by Hernández-Ortiz et al. [21–23]: Andean, Brazilian-1, Brazilian-2, Brazilian-3, Ecuadorian, Mexican, Peruvian, and Venezuelan. The fruit fly *A. fraterculus* feeds on at least 160 crop species [24,25]. In Brazil, it is considered one of the primary pests affecting apple (*Malus domestica* Borkh.), peach [*Prunus persica* (L.) Batsch], and guava (*Psidium guajava* L.) [25,26]. In Colombia, *A. fraterculus* is a quarantine species reported feeding on coffee berries (*Coffea arabica* L.) in the Andes mountains at up to 1.100 MASL (meters above sea level) [27–29]. In Peru, this species is common in mango crops (*Mangifera indica* L.) [30,31]. In Mexico, this pest is particularly problematic for mango crops, which is a major agricultural product in many areas of the country [32].

One of the most common methods for controlling *Anastrepha* fruit flies involves classical, augmentative, and conservative biological control techniques, utilizing native and exotic parasitoids [32–34]. Some of the more prevalent parasitoid species of *Anastrepha* spp. are *Doryctobracon areolatus* (Szépligeti, 1911) (Hymenoptera: Braconidae), native to South America, and *Diachasmimorpha longicaudata* (Ashmead, 1905) (Hymenoptera: Braconidae), native to Southeast Asia [35,36]. The parasitism rates of *D. areolatus* on *A. fraterculus* varied from 41.6% to 68.6% (53.5% on average), demonstrating its potential in biological control programs [37,38]. Similarly, the parasitism rates of *D. longicaudata* on *A. fraterculus* ranged from 12.9% to 62.9% when reared on citrus and guava fruits in laboratory conditions and 35.5% in artificial diet [39,40]. Recently, the Brazilian Ministry of Agriculture and Livestock (MAPA) has registered *D. areolatus* as a fruit fly biocontrol agent for *Anastrepha* spp. [41].

The parasitoid *D. longicaudata* has been introduced to different countries in the Americas, including Colombia, Venezuela, Brazil, Costa Rica, Guatemala, El Salvador, Mexico, Nicaragua, Trinidad, and the United States of America (Florida) as a biological control agent of many tephritids [42,43]. This parasitoid has been found parasitizing *A. fraterculus* in Costa Rica, Mexico, Nicaragua, Guatemala, El Salvador, Trinidad, Colombia, and Venezuela [42]. Furthermore, augmentative releases of *D. longicaudata* in Mexico have resulted in up to 50% parasitism of *Ceratitis capitata* (Wiedemann, 1824) [44]. This parasitoid was also introduced to Argentina in the 1960s and 1990s to control *C. capitata* and *A. fraterculus* [45]. In Brazil, it was introduced in 1994 and officially registered as a biocontrol agent of tephritids in 2018 [33,46]. However, despite some releases in the states of Bahia and São Paulo in Brazil, where the parasitism rates reached values of 19.8% in citrus orchards [47], the parasitoid has not been successfully established speculatively due to intraspecific competition with native parasitoid species. *D. longicaudata* also failed to establish in the state of Rio Grande do Sul due to low temperatures in the region [43,48]. These studies suggest that *D. areolatus* and *D. longicaudata* require different ecological conditions for their successful establishment.

This study predicts the potential geographic distribution of the fruit fly *A. fraterculus* and its parasitoids *D. areolatus* and *D. longicaudata* across the Americas under current and future climate conditions. Additionally, this study forecasts potential population range shifts using a comprehensive set of climate projection models, considering two shared socioeconomic pathways of gas emissions (SSP) and two distinct periods of time. We tested the following hypotheses: 1) the suitable areas for the establishment of the fruit fly *A. fraterculus* will increase in the future; 2) the suitable areas for the establishment of the parasitoids *D. longicaudata* and *D. areolatus* will not increase in the future; and 3) there will be less suitable areas for the introduced parasitoid *D. longicaudata* than for the native parasitoid *D. areolatus* in future climate change scenarios. The results of this study contribute to the development of pest management programs for *A. fraterculus* by determining potential areas for the establishment of this pest and two of its parasitoid species in the Americas.

## Materials and methods

### Species record occurrences

The geographical coordinates of species occurrences used to build the SDM analysis for *A. fraterculus*, *D. longicaudata,* and *D. areolatus* were obtained using the "occ" function from the "spocc" RStudio package [49]. The occurrence records of these species were sourced from the following databases: The Global Biodiversity Information Facility (GBIF; https://www.gbif.org), The Vertebrate Network (VerNet, http://vertnet.org/index.html), iNaturalist (https://www.inaturalist.org/), and The Integrated Digitized Biocollections (iDigBio). Additional occurrences of *D. areolatus* were extracted from the records reported by Moreira et al. [50]. After filtering the data to remove duplicates, we obtained a total of 193 records for *A. fraterculus*, 21 for *D. longicaudata*, and 205 for *D. areolatus*. The georeferenced occurrences for each species were then compiled and saved in a ".csv" file for subsequent analysis.

Both occurrences and absences are usually necessary to build SDMs. Nevertheless, due to the lack of true absence data in the databases, "pseudo-absence" locations were generated randomly to characterize the environmental background of the study area. To do this, we used the function "randomPoints" from the "dismo" package [51] in R statistical

computing software v4.3.2; [52]. These pseudo-absences are locations where it is assumed that there is no record or presence of the species, thus providing a point of comparison from those locations where the species has been recorded. In this study, a ratio of 2:1 pseudo-absence to presence locations was generated for each species, reducing over-fitting and maintaining the capacity of the model to discriminate between areas of presence and absence [53].

## Environmental variables

SDMs use data on environmental conditions in which a species is known to exist to predict suitable habitats for that species. Environmental data for current and future climate conditions in the Americas were obtained from the WorldClim database (Global Climate Data; available at: https://www.worldclim.org/data/index.html) [54]. These data consist of 19 bioclimatic variables derived from temperature and precipitation in "raster" format (Table 1), at a spatial resolution of 2.5 arc-minutes (approximately 4.5 km).

Bioclimatic variables used for SDMs (extracted from www.worldclim.org/data/bioclim.html**).**

The current environmental conditions were downloaded from the WorldClim database version 2.1 and correspond to historical records from 1970 to 2000 [54]. The environmental conditions for future climate scenarios were obtained from 26 climatic projections or Global Climate Models (GCM) from the Coupled Model Intercomparison Project Phase 6 (CMIP6; available at https://www.worldclim.org/data/cmip6/cmip6climate.html, [55]) (S1 Table). These projections consist of climatic predictions made by various modeling groups based on $CO_2$ emissions and mitigation effort scenarios up to the year 2100, called Shared Socioeconomic Pathways (SSPs). We selected two SSPs at two time periods, from 2021 to 2040 and from 2041 to 2060: (a) SSP2–4.5, which represents a stabilization scenario that assumes the implementation of climatic policies to mitigate greenhouse gas emissions [56]; and (b) SSP5–8.5, which corresponds to high greenhouse gas emissions pathway and is considered a baseline scenario without any specific targets for climate mitigation [57].

**Table 1. Bioclimatic variables.**

| Code | Variables |
|---|---|
| BIO1 | Annual Mean Temperature |
| BIO2 | Mean Diurnal Range (Mean of monthly (max temp – min temp)) |
| BIO3 | Isothermality ((BIO2/BIO7) ×100) |
| BIO4 | Temperature Seasonality (standard deviation of monthly mean temperatures ×100) |
| BIO5 | Max Temperature of Warmest Month |
| BIO6 | Min Temperature of Coldest Month |
| BIO7 | Temperature Annual Range (BIO5-BIO6) |
| BIO8 | Mean Temperature of Wettest Quarter |
| BIO9 | Mean Temperature of Driest Quarter |
| BIO10 | Mean Temperature of Warmest Quarter |
| BIO11 | Mean Temperature of Coldest Quarter |
| BIO12 | Annual Precipitation |
| BIO13 | Precipitation of Wettest Month |
| BIO14 | Precipitation of Driest Month |
| BIO15 | Precipitation Seasonality ((standard deviation of monthly precipitation/mean monthly precipitation) x 100) |
| BIO16 | Precipitation of Wettest Quarter |
| BIO17 | Precipitation of Driest Quarter |
| BIO18 | Precipitation of Warmest Quarter |
| BIO19 | Precipitation of Coldest Quarter |

## Suppression of collinearity in predictor variables

Collinearity, also known as multicollinearity, refers to the strong interdependence among explanatory variables typically observed in regression models. This phenomenon can lead to two primary issues: inflated estimates of variable effects and uncertainty in model extrapolation [58]. The most common approach for addressing collinearity issues is a priori variable selection, which involves either excluding the most correlated variables or combining them into new explanatory terms [59]. For this purpose, each species occurrence datum was projected into the current environmental conditions database, and the values of each bioclimatic variable at those points were extracted. Subsequently, the Variance Inflation Factor (VIF) was calculated using the "vifstep" function from the "usdm" package [60]. This test excluded the highly correlated variables with VIF values >10; the remaining variables were used to run the SDMs projections.

## Modeling and projections

Random Forest is a powerful machine-learning method for building SDMs and training models with small sample sizes [61–63]. This method consists of hundreds of trees constructed using random samples from the databases of occurrences/pseudo-absences and selected bioclimatic variables (predictors). Each tree is built from a bootstrap sample, which is a random sample with replacement from the original dataset. At each node of the tree, the best predictor is selected from a small, randomly chosen subset of the predictor variable pool. This process helps introduce randomness and reduce overfitting of the model [64]. The final prediction is then made by aggregating the predictions of all individual trees. This ensemble approach tends to improve the robustness and generalization performance of the model compared to the use of a single decision tree [65].

Random Forest involves several parameters that control the structure of each tree, the overall structure and size of the forest, and its randomness properties. For example, the number of randomly drawn candidate variables (mtry), the node size, and the number of trees (ntree). The optimal selection of these parameters significantly influences model prediction accuracy [66]. To determine the best values for these parameters, strategies such as "Tuning" have been developed [67]. In this context, parameter selection was performed using the "train" function from the "caret" package [68]. For all predictions made for *A. fraterculus* and *D. areolatus*, a ntree of 2000, a mtry of 3, and a node size of 2 with 25 replications were selected. However, for all predictions involving *D. longicaudata*, a ntree of 2500, a mtry of 3, and a node size of 2, with 10 replications were selected. Subsequently, all the models were adjusted using 75% of the occurrence data to train the model and the remaining 25% for testing purposes. An ensemble was created using all replications for the current projection. For future projections, an ensemble was formed by combining all 25 Global Climate Models for each climatic period (2021–2040 and 2041–2060) and Representative Concentration Pathway (SSP).

SDMs using Random Forest usually provide numerical predictions in the range of 0–1. However, to make the data more practical, it is essential to establish a threshold for interpreting the predictions using levels of suitability and transforming them into binary data indicating species presence or absence. Suitability scores higher than the threshold value are considered suitable areas for the establishment of a given species. In this study, threshold values were obtained for each insect species by extracting all the Receiver Operating Characteristic (ROC) curve data using the function "roc" of the "pROC" package [69]. Subsequently, the threshold was determined as the mean of all the threshold values in each replication for each species (25 for *A. fraterculus* and *D. areolatus*, and 10 for *D. longicaudata*). We used four suitability levels represented on a color gradient from green to red on the plotted maps: a) highly suitable areas depicted in green color correspond to suitability values close to 1; b) moderately suitable areas illustrated in yellow designate suitability values close to 0.5; c) low suitable areas depicted in red show suitability values close to the threshold for each species (0.08 for *A. fraterculus*, 0.31 for *D. areolatus* and 0.15 for *D. longicaudata*); and d) unsuitable areas represented by the absence of color correspond to values below the threshold and close to 0.

Lastly, the suitability values for each predicted scenario and period were converted into binary values using a species-specific threshold. This transformation was conducted for two key purposes: 1) to quantify the suitable and

unsuitable areas in square kilometers for all species, facilitating the assessment of expansion areas across different countries in the Americas; 2) to generate maps illustrating the overlap of suitable areas between each parasitoid and the pest *A. fraterculus.* For the first one, we utilized the "expanse" function from the "terra" package [70]. The resulting data were subsequently saved in a ".csv" file. To aid interpretation, the top 10 countries with the largest suitable areas for the fruit fly *A. fraterculus* were selected. The same countries were also chosen for quantifying the suitability areas of the parasitoids *D. areolatus* and *D. longicaudata*, and the data were visualized using a bar plot.

All the maps generated illustrating the suitable areas for each species, as well as the overlap of suitable areas between the parasitoids and pest, were generated using the function "ne_countries" of the "rnaturalearth" RStudio package [71].

## Model performance

To assess the performance of the models, we used the ROC curve analysis, specifically examining the area under the curve (AUC) [72]. The AUC value is a widely accepted metric for evaluating the accuracy of SDMs. Generally, AUC values below 0.7 indicate poor performance, those between 0.7 and 0.9 suggest moderate performance, and values exceeding 0.9 indicate good performance [73]. For each species, the relative importance of the selected bioclimatic variables was determined through analysis of multicollinearity and AUC metrics. This comprehensive approach allowed us to gauge both the predictive power of the models and the significance of individual variables in the SDMs.

## Results

The predicted SDMs constructed for *A. fraterculus*, *D. areolatus*, and *D. longicaudata*, had good performance with AUC values of 0.995±0.005, 0.963±0.034, and 0.9583±0.1317, respectively. These models indicate that suitable areas for the pest *A. fraterculus* and the parasitoids *D. areolatus* and *D. longicaudata*, were predicted to increase in the Americas in all the scenarios and periods evaluated (Table 2). The species with the largest suitable areas in each prediction (SSPs scenario and periods) was *D. longicaudata*, estimated to occupy up to 38.06% of the America's territory, followed by *A. fraterculus* with 36.17%, and *D. areolatus* with 29.66% (Table 2). The predicted occupancy areas had no marked differences between the climatic scenarios analyzed (Table 2).

Among the bioclimatic variables analyzed, isothermality (BIO3) had the highest contribution to the *A. fraterculus* SDM with 20.51%, followed by the mean temperature of the driest quarter (BIO9) with a contribution of 14.22%, and the precipitation of the warmest quarter (BIO18) with 13.52% (Table 3). For *D. areolatus* SDM, the bioclimatic variable with the major contribution was the annual temperature range (BIO7) with 25.35%, followed by isothermality (BIO3) with a contribution of 16.20%, and precipitation of the wettest month (BIO13) with 12.05%. For *D. longicaudata*, two main variables contributed more than 70% to the SDM model: the average temperature of the coldest quarter (BIO11) with 51.02% and the precipitation of the wettest quarter (BIO18) with 27.33% (Table 3).

### The *Anastrepha fraterculus* SDM

Our models predict that *A. fraterculus* has an extensive suitable area for establishment in South America, with Brazil exhibiting the largest extent (up to 5 million km²), followed by Peru (up to 1 million km²) and Colombia (up to 0.8 million km²) (Fig

**Table 2. Percentage of predicted suitable areas for the establishment of the pest *Anastrepha fraterculus*, and the parasitoids *Doryctobracon areolatus* and *Diachasmimorpha longicaudata* in the Americas in two Shared Socioeconomic Pathways (SSPs) at two periods.**

| Species | | SSP2–4.5 | | SSP5–8.5 | |
|---|---|---|---|---|---|
| | Current | 2021-2040 | 2041-2060 | 2021-2040 | 2041-2060 |
| *Anastrepha fraterculus* | 31.39% | 34.56% | 35.55% | 34.58% | 36.17% |
| *Diachasmimorpha longicaudata* | 35.07% | 36.66% | 37.42% | 36.79% | 38.06% |
| *Doryctobracon areolatus* | 21.53% | 26.25% | 28.37% | 26.59% | 29.66% |

**Table 3. The relative contribution of the bioclimatic variables for Random Forest models built for *Anastrepha fraterculus*, *Doryctobracon areolatus*, and *Diachasmimorpha longicaudata*.**

| Species | Codes | Bioclimatic variables | Contribution (%) |
|---|---|---|---|
| *Anastrepha fraterculus* | BIO2 | Mean Diurnal Range | 10.02 |
| | BIO3 | Isothermality | 20.51 |
| | BIO8 | Mean Temperature of Wettest Quarter | 11.02 |
| | BIO9 | Mean Temperature of Driest Quarter | 14.22 |
| | BIO13 | Precipitation of Wettest Month | 7.70 |
| | BIO15 | Precipitation Seasonality | 11.54 |
| | BIO18 | Precipitation of Warmest Quarter | 13.52 |
| | BIO19 | Precipitation of Coldest Quarter | 11.45 |
| *Doryctobracon areolatus* | BIO3 | Isothermality | 16.20 |
| | BIO7 | Temperature Annual Range | 25.35 |
| | BIO8 | Mean Temperature of Wettest Quarter | 8.22 |
| | BIO10 | Mean Temperature of Warmest Quarter | 9.88 |
| | BIO13 | Precipitation of Wettest Month | 12.05 |
| | BIO14 | Precipitation of Driest Month | 6.04 |
| | BIO15 | Precipitation Seasonality | 4.57 |
| | BIO18 | Precipitation of Warmest Quarter | 10.79 |
| | BIO19 | Precipitation of Coldest Quarter | 6.89 |
| *Diachasmimorpha longicaudata* | BIO 11 | Mean Temperature of Coldest Quarter | 52.02 |
| | BIO15 | Precipitation Seasonality | 4.36 |
| | BIO16 | Precipitation of Wettest Quarter | 27.33 |
| | BIO18 | Precipitation of Warmest Quarter | 11.83 |
| | BIO19 | Precipitation of Coldest Quarter | 4.46 |

1). In Brazil, the projected suitable areas for the current time occupy 61% of the national territory, with a potential increase of up to 75% under future climate scenarios (S1 Table). These predicted suitable areas were predominant in the southern states (Rio Grande do Sul, Santa Catarina, Paraná), southeastern states (São Paulo, Minas Gerais, Rio de Janeiro, Espírito Santo), and parts of Bahia State in Brazil (Fig 2). In Peru, the projected suitable areas for *A. fraterculus* extend up to 98% of the country, with minimal variation in future projections (S1 Table); these suitable areas were predicted in the Andes mountains and adjacent valleys (Fig 2). Conversely, Colombia is projected to experience a significant expansion of suitable areas, from 70% to 99% of its territory under future scenarios (Fig 1), with new suitable areas emerging in the Plains (East Colombia) and Amazon (South Colombia) regions, showing moderate to low suitability for *A. fraterculus* (Fig 2).

In North America, current time predictions indicate suitable areas for *A. fraterculus* along the east coast of the United States (0.9 million km²) and in Mexico (0.8 million km²) (Fig 1), particularly in the coastal states of Yucatán, Campeche, Chiapas, Oaxaca, and Tabasco. In the United States, the suitable areas for this pest are predicted to increase in future climate scenarios by 12% of its territory, concentrated in the states of Ohio, Indiana, and Kentucky (Fig 2). In contrast, Mexico was projected to experience a slight decline of suitable areas for *A. fraterculus* from 39% to 36% of its territory in future scenarios (S2 Table). The Mesoamerican countries such as Panama, Costa Rica, Guatemala, and Honduras, display low to high suitability across nearly all their territories in both current and future projections, with minimal temporal variation (Fig 2).

### The fruit fly parasitoid, *Doryctobracon areolatus* SDM

Moderate to high suitable areas for *D. areolatus* were identified across South America, particularly along the western and coastal regions of Brazil, as well as in northern Venezuela, Paraguay, Colombia, and Bolivia (Fig 3). In contrast, areas

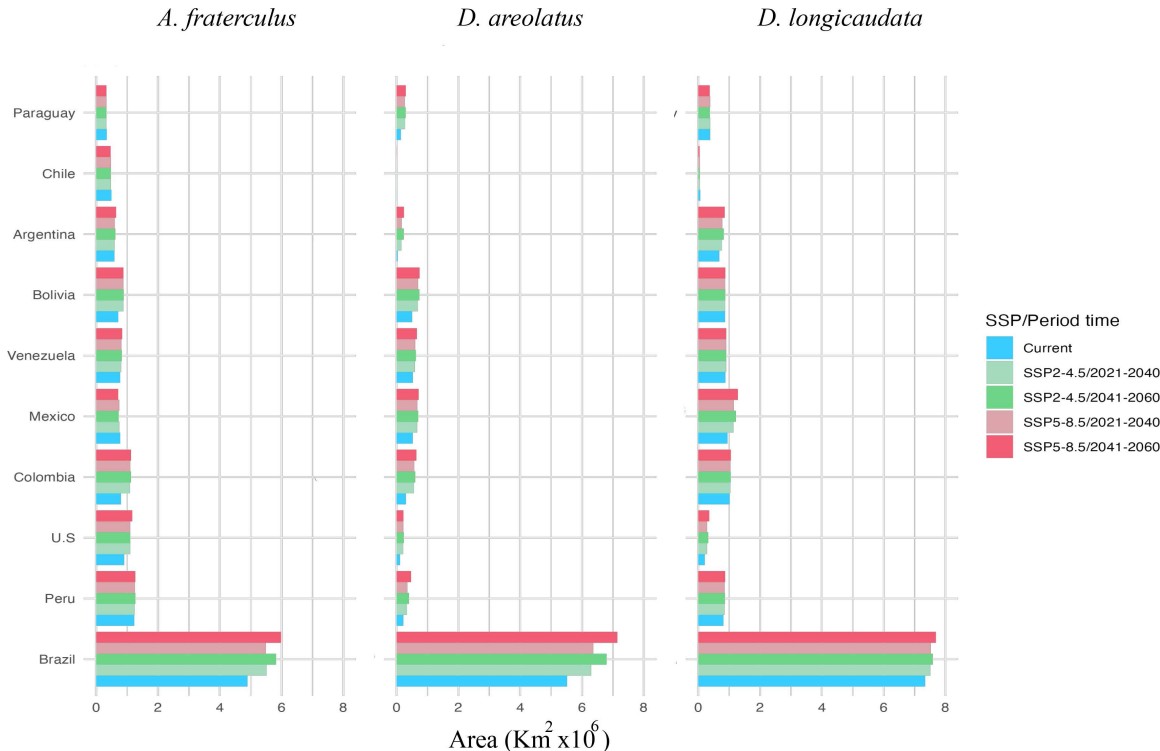

**Fig 1. Top 10 countries with the largest suitable areas (Km²) for *Anastrepha fraterculus*.** Predicted suitable areas for the parasitoids *Diachasmimorpha longicaudata* and *Doryctobracon areolatus* and the pest *A. fraterculus* in current and future time predictions [(SSP2-4.5, SSP5-8.5), (2021-2040, 2041-2060)].

with low suitability were predicted in the Amazon region and some areas in the Andes mountains, spanning Chile, Peru, Ecuador, and Colombia. Additionally, some areas were classified as low to moderately suitable for the establishment of *D. areolatus* in Central and North America (Fig 3).

In Brazil, current projections estimate that 69% of the territory is suitable for the establishment of *D. areolatus*, with future climate scenarios predicting an increase of up to 89% (Fig 1). In Peru, suitability is projected to increase from 16% to 36%, and in Colombia, from 26% to 56% (Fig 1), mainly in the Plains and Amazon regions (Fig 3). In Mexico, the current suitable areas predicted for *D. areolatus* are concentrated in the southeast, southwest, and central-south (Fig 3), with future projections suggesting a 10% increase (Fig 1; S2 Table). In the United States, suitable areas were consistently identified along the Southeast Coast, particularly in Florida, under all projections (Fig 3; S2 Table).

Overlapping suitable areas between *A. fraterculus* and *D. areolatus* were projected in the south and littoral region of Brazil, in the north of Colombia and Venezuela, and in the majority of Mesoamerica (Fig 4). In Mexico, overlapping suitable areas were more predominant in the Yucatan peninsula and in the south part of the country (Fig 4). In contrast, in the United States of America, overlapping areas were restricted to the south of Florida (Fig 4). Future projections show an increase of overlapping suitable areas between *A. fraterculus* and *D. areolatus* in almost all the Brazilian territory, except in some areas in the Amazon region and the center region of the country (Fig 4; S1 Fig). However, in Colombia, common suitable areas for *A. fraterculus* and *D. areolatus*, were predicted in the Amazon and the Plains (Fig 4). In contrast, in Mexico and the United States of America, common suitable areas for the pest and the parasitoid had no significant change in future predictions (Fig 4; S1 Fig).

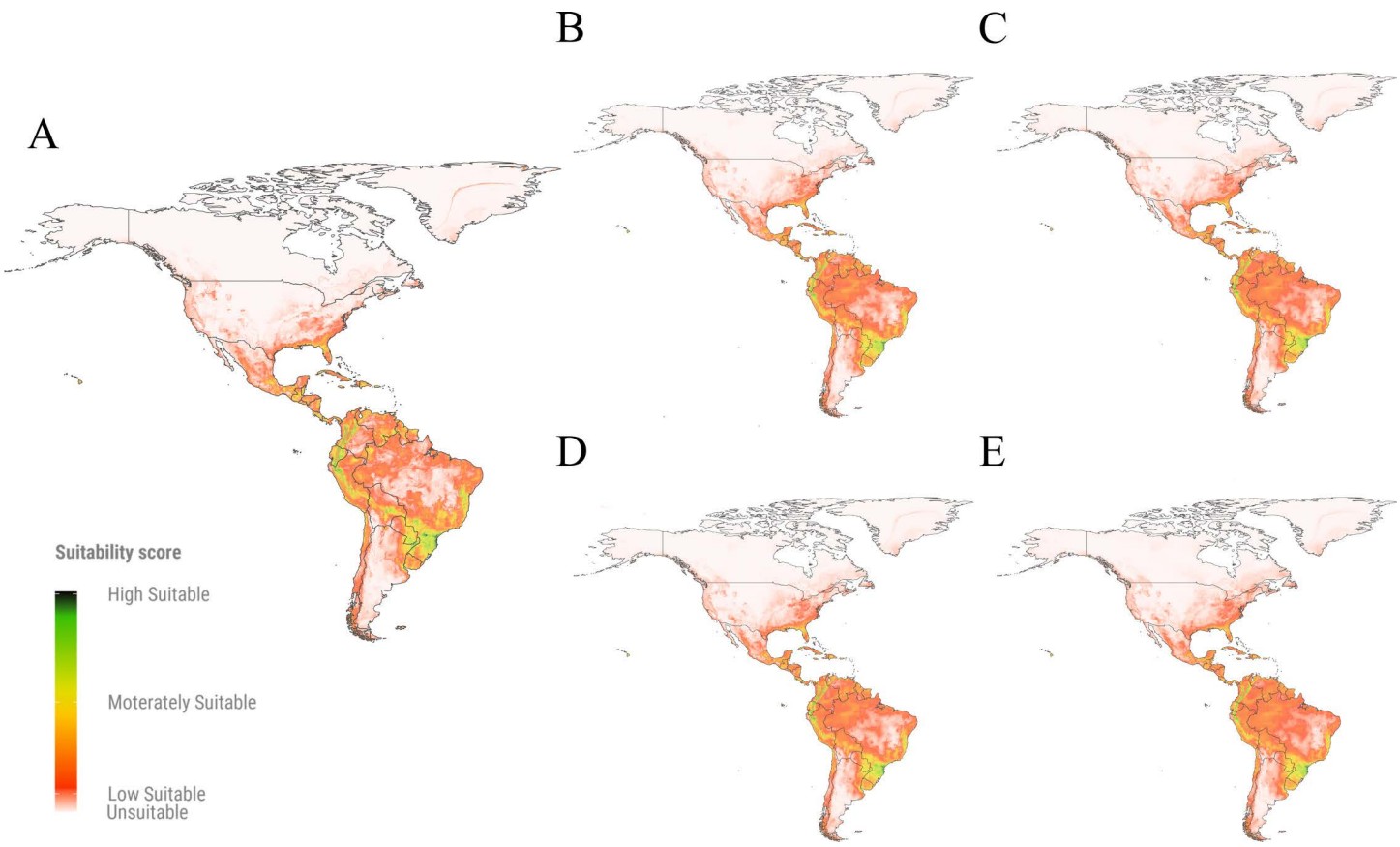

**Fig 2. Suitable areas predicted for *Anastrepha fraterculus* in the Americas. (A).** Suitable areas predicted in the current time period*; (B-E).* Suitable areas predicted under two future climate change scenarios (SSP2-4.5 and SSP5-8.5) (periods 2021-2040 and 2041-2060); (B) scenario SSP2-4.5 time period 2021-2060; (C) scenario SSP2-4.5 time period 2041-2060; (D) scenario SSP5-8.5 time period 2021-2040; (E) scenario SSP5-8.5 time period 2041-2060. The maps show suitability values in a green to-red color where: high suitable areas depicted in green color correspond to suitability values close to 1; moderately suitable areas illustrated in yellow designate suitability values close to 0.5; low suitable areas depicted in red show suitability values close to the threshold of the species (0.08); unsuitable areas represented by the absence of color correspond to values below the threshold and close to 0.

### The fruit fly parasitoid*, Diachasmimorpha longicaudata* SDM

Our predictions showed that suitable areas for *D. longicaudata* occupy 35.07% of the Americas territory (Table 2). Future projections indicate a slight expansion of suitable areas of up to 38.06% in the Americas (Table 2). Highly to moderately suitable areas for *D. longicaudata* were projected in the Center-West and North of Brazil, with an increase in suitable areas of 14% in its territory in future predictions (Fig 5, S2 Table). In Peru, suitable areas for the current period were predicted to cover 63% of the national territory, which could increase to 67% in future scenarios (Fig 1; S2 Table). In Colombia, future projections indicate that up to 90% of the country will be suitable for *D. longicaudata* (Fig 1; S2 Table), with moderate to high suitability across most regions, except for the Andes mountains, which remain unsuitable in future predictions (Fig 5.). In the United States, suitable areas are limited to central and southern Florida, with no significant expansion observed in future scenarios (Fig 5). Across Mesoamerica, future projections suggest moderate suitability for *D. longicaudata* along coastal areas and low suitability in the Northeast, lowlands, and central regions of Mexico (Fig 5). Suitable areas in Mexico are expected to increase 17% in future projections.

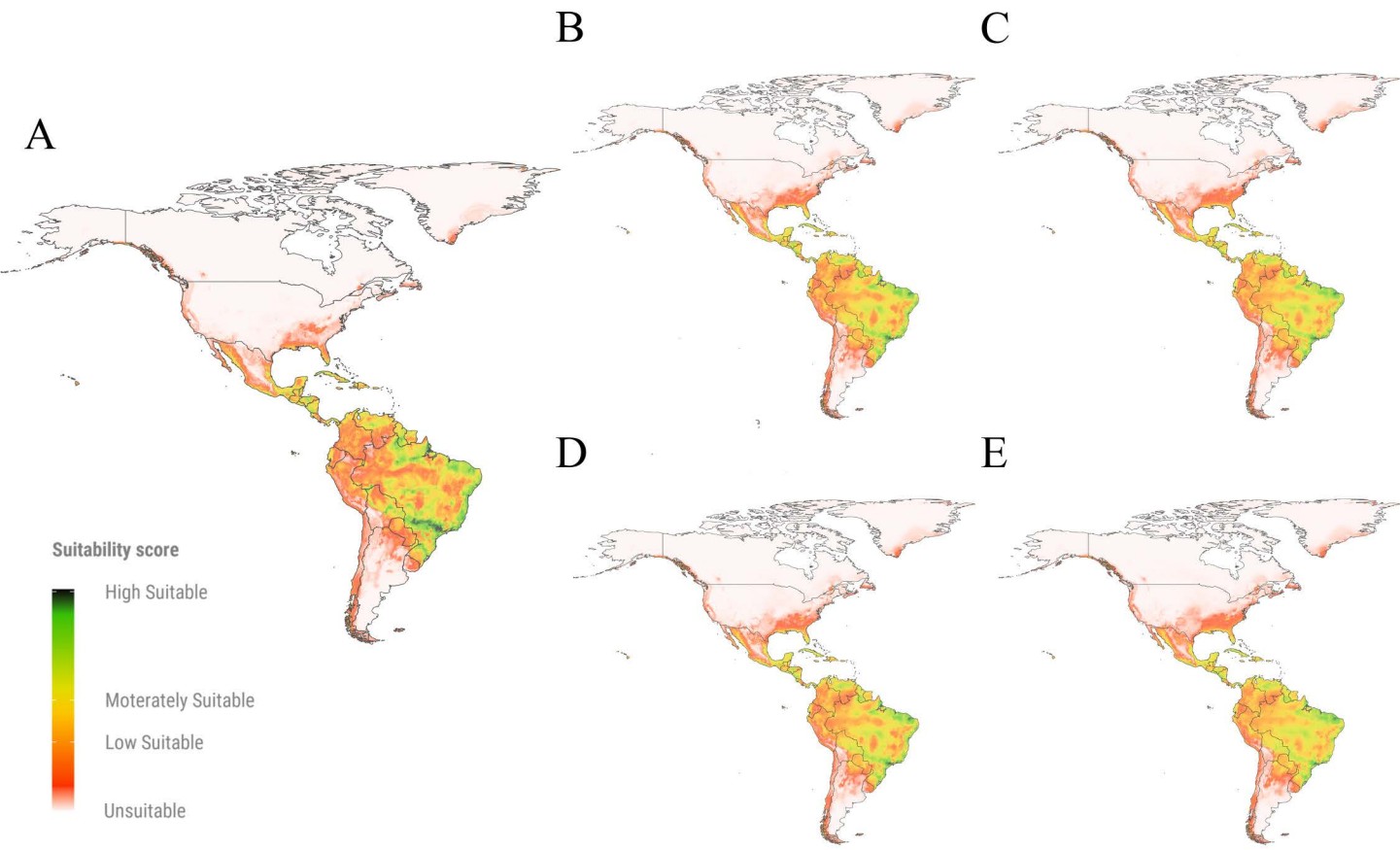

**Fig 3. Suitable areas predicted for *Doryctobracon areolatus* in the Americas. (A)** Suitable areas predicted in the current time period; **(B-E)**. Suitable areas predicted under two future climate change scenarios (SSP2-4.5 and SSP5-8.5) and periods (2021-2040 and 2041-2060); (B) scenario SSP2-4.5 time period 2021-2060; (C) scenario SSP2-4.5 time period 2041-2060; (D) scenario SSP5-8.5 time period 2021-2040; (E) scenario SSP5-8.5 time period 2041-2060. The maps show suitability values in a green to-red color where: high suitable areas depicted in green color correspond to suitability values close to 1; moderately suitable areas illustrated in yellow designate suitability values close to 0.5; low suitable areas depicted in red show suitability values close to the threshold of the species (0.31); unsuitable areas represented by the absence of color correspond to values below the threshold and close to 0.

Notably, there is a substantial overlap of suitable areas for *D. longicaudata* and *A. fraterculus*, particularly in South America and Mesoamerica (Fig 6; S2 Fig). Current predictions showed suitable overlapping areas across the Amazon region in Brazil, Peru, and Colombia; these areas are predicted to have a significant expansion in future projections (Fig 6; S2 Fig). In Mexico, overlapping suitable areas for the establishment of *D. longicaudata* and *A. fraterculus* were predicted along the coast in both current and future predictions (Fig 6; S2 Fig). In the United States of America, overlapping suitable areas for the parasitoid and the pest were predicted only in the state of Florida (Fig 6; S2 Fig).

## Discussion

### *Anastrepha fraterculus* is predicted to expand to new territories in the Americas under future climate change scenarios

Our models predict a reduced ability of *A. fraterculus* to establish in warmer, dry regions, and a high ability to establish in temperate areas (Fig 2). These predictions agree with previous laboratory bioassays that found a reduced emergence of

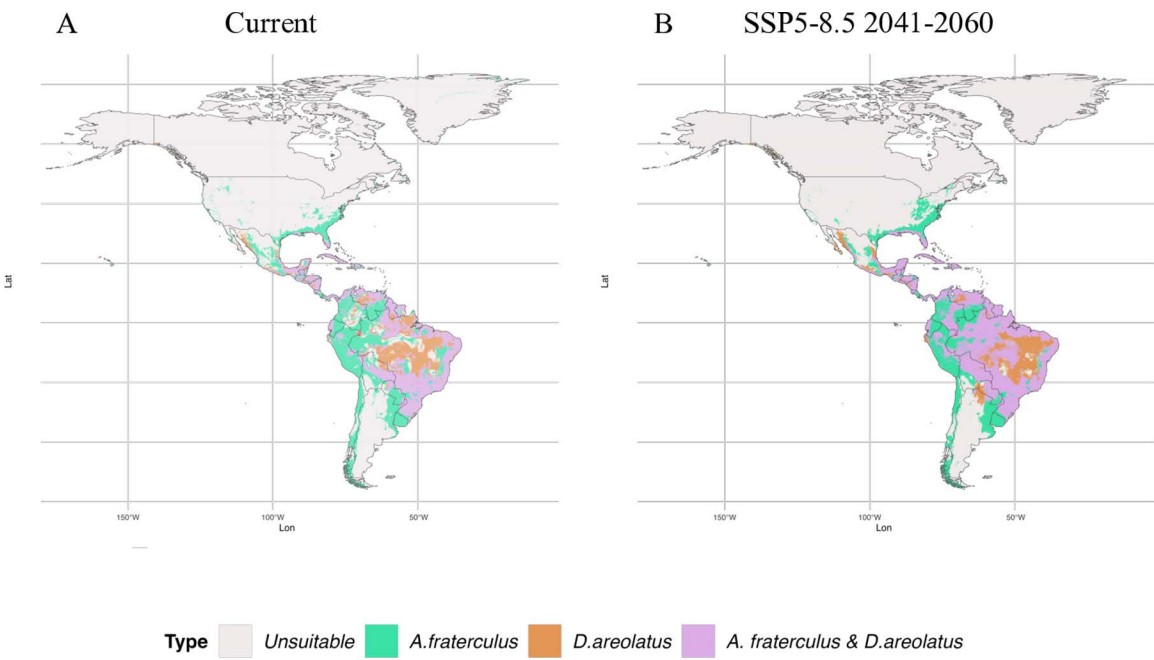

**Fig 4. Overlapping suitable areas predicted for *Doryctobracon areolatus* and *Anastrepha fraterculu*s in the Americas. (A)** Overlapped suitable areas predicted in the current time period; **(B)** Overlapped suitable areas predicted in the future 2041-2060 scenario SSP5-8.5.

*A. fraterculus* adults at temperatures exceeding 30°C, whereas the emergence rate exceeded 70% at 15 °C [74]. Similarly, Santos [75] concluded that water stress due to scarcity can cause mortality of *A. fraterculus* under warmer conditions. Our results are comparable with those reported for *Anastrepha grandis* SDM, where the precipitation of the warmest quarter (BIO10) was the most influential variable, suggesting that extreme drought events may drive the future distribution of this species [76]. Also, in our predictions, regions with temperate climates showed moderate and high suitability for the establishment of *A. fraterculus*, whereas low-suitable and unsuitable areas include some tropical climate regions. Based on this, we suggest that temperate climate areas with high humidity seem to be more suitable for the establishment of this species at the current time and in the future. Indeed, the spread of *A. fraterculus* in temperate climate regions due to climate change has already been reported in the field [77].

In Brazil, temperate climates are predominant in the southern and southeastern regions and are characterized by hot summers, no dry seasons, and cooler winters [78]. However, our study found suitable areas for the establishment of *A. fraterculus* in the coastal region characterized by high temperature, low altitude, and high humidity [79]. Our projections concur with previous studies reporting the presence of different *A. fraterculus* morphotypes in these regions. For example, the Brazilian-2 and Brazilian-3 morphotypes seem to be more adapted to coastal areas, whereas the Brazilian-1 morphotype seems to thrive in temperate areas [22,80,81]. Our model predicted areas with low or no suitability for *A. fraterculus* in the Brazilian Amazon region and central-western states. These regions experience tropical climates, characterized by dry winters and summers, whereas the Brazilian Amazon has predominantly tropical monsoon and tropical rainforest climates [78,82]. The low suitability predicted in these areas might be the result of not conducting separate analyses for each *A. fraterculus* morphotype. SDMs developed by Selivon et al. [81] for each *A. fraterculus* morphotype in Brazil, found suitable areas only in the southeast and south of the country. On the other hand, populations of *A. fraterculus* in Meso-america, known as the 'Mexican morphotype', were found at low elevations (30–1,400m) and thrive in both temperate and tropical climates [22,23]. Therefore, our results of the expansion of the suitable regions for *A. fraterculus* in the Brazilian

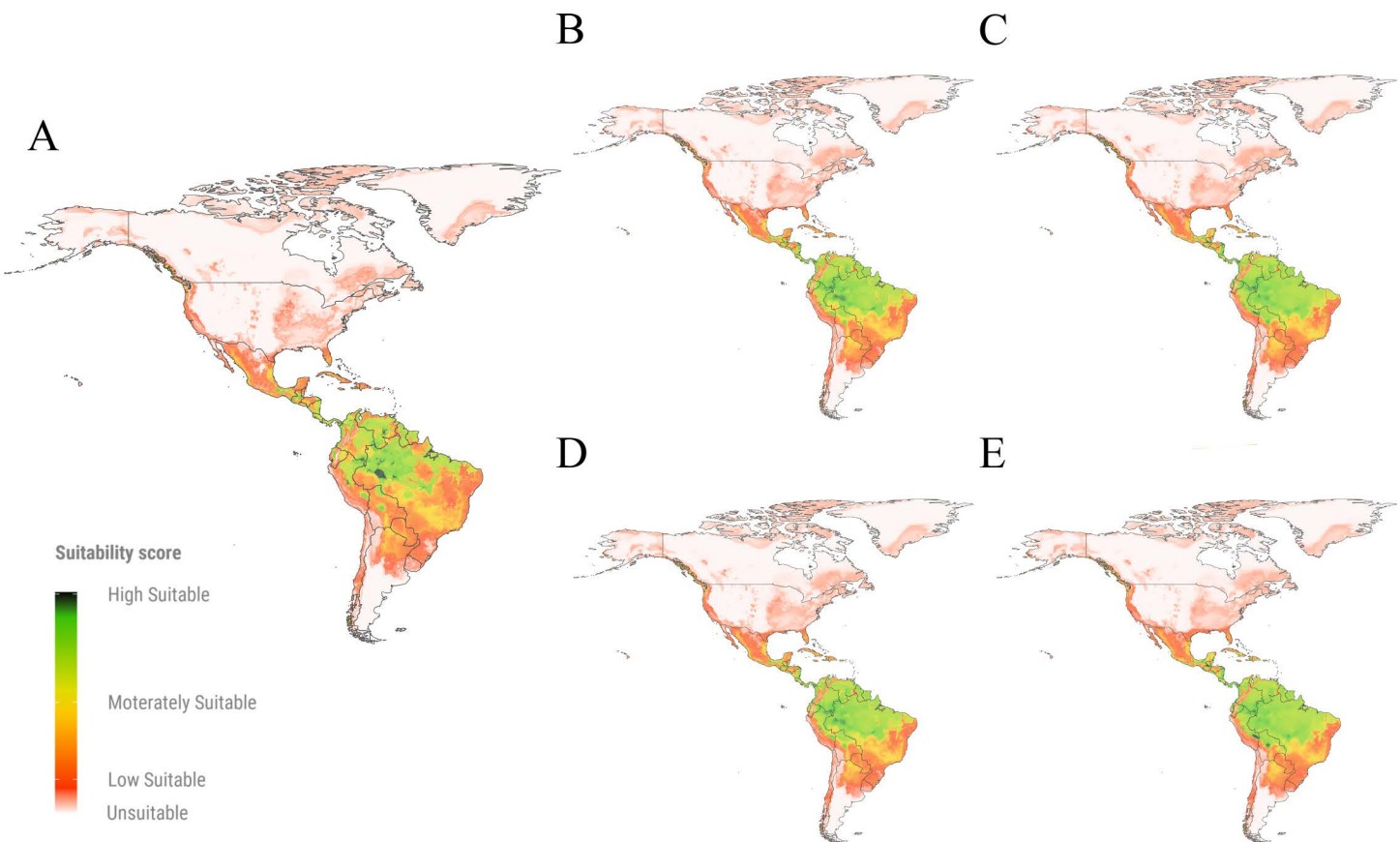

**Fig 5. Suitable areas predicted for *Diachasmimorpha longicaudata* in the Americas (A) Suitable areas predicted in the current time period; (B-E).** Suitable areas predicted under two future climate change scenarios (SSP2-4.5 and SSP5-8.5) and periods (2021-2040 and 2041-2060); (B) scenario SSP2-4.5 time period 2021-2060; (C) scenario SSP2-4.5 time period 2041-2060; (D) scenario SSP5-8.5 time period 2021-2040; (E) scenario SSP5-8.5 time period 2041-2060. The maps show suitability values in a green to-red color where: high suitable areas depicted in green color correspond to suitability values close to 1; moderately suitable areas illustrated in yellow designate suitability values close to 0.5; low suitable areas depicted in red show suitability values close to the species threshold (0.15); unsuitable areas represented by the absence of color correspond to values below the threshold and close to 0.

Amazon are for the species without differentiation between its morphotypes. However, the continued presence of *A. fraterculus* in future scenarios in the Brazilian southern and southeastern regions could pose a risk for crops, such as apples and peaches in these regions [26,83].

The Peruvian mountains, predicted by our study as suitable for *A. fraterculus*, are characterized by two types of climates, temperate and arid [78]. A study conducted by Ramos Peña et al. [84] in the Abancay Valley in Apurimac, Peru, found an increase in fruit fly populations during the rainy season and the transitions to the dry season. The authors also suggested that low temperatures and the availability of suitable hosts increase fruit fly populations. Studies conducted in guava orchards in Oxapampa, Peru, revealed an increase in the abundance and uniform distribution of *A. fraterculus* at higher elevations. Moreover, these trends appear to be influenced by both seasonality and altitude [85]. In Colombia, the presence of *A. fraterculus* in coffee crops in the Andes mountains was previously reported by Jaramillo et al. [28], representing 99.9% of all specimens of fruit flies in coffee berries; however, its effects on fruit development and coffee quality are unknown. Additionally, at least 37 other hosts for *A. fraterculus* have been reported in Colombia [86]. Some of these hosts are important crops in the Plains and Amazon regions and will be at risk in the future, such as mango [87,88],

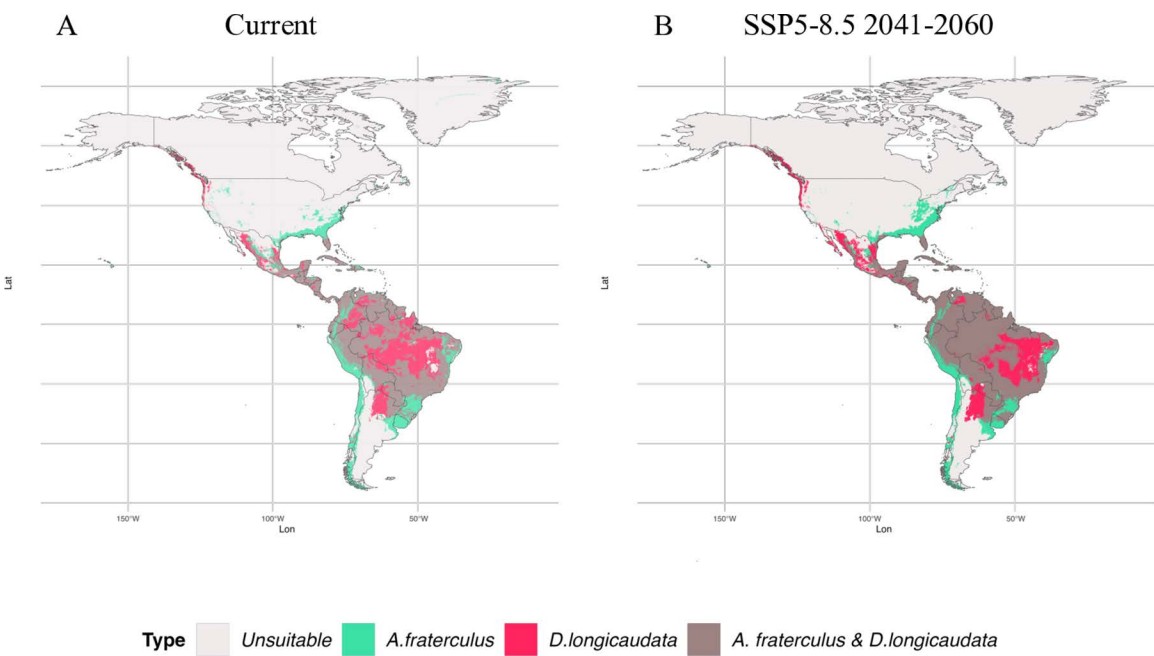

**Fig 6. Predicted overlapping suitable areas for the establishment of *Diachasmimorpha longicaudata* and *Anastrepha fraterculus* in the Americas. (A)** Overlapping suitable areas predicted in the current time period; **(B)** Overlapping suitable areas predicted in the future 2041-2060 scenario SSP5-8.5.

soursop (*Annona muricata* L.), papaya (*Carica papaya* L.) [87], and tangerine (*Citrus reticulata* Blanco) [27]. Given the potential increase of suitable areas in the Plains and Amazon regions of Colombia in the coming years, efforts to monitor the dispersion of *A. fraterculus* are highly recommended.

Our results show suitable areas for *A. fraterculus* in the Yucatan peninsula in Mexico, where this pest is currently present [23]. The loss of suitable areas predicted by our models in Mexico can be explained by a reduction of temperate climate regions, transitioning to arid climates in the future [78]. This suggests that arid climates are not optimal for the establishment of the species. Indeed, states such as Chihuahua, Durango, Coahuila, Nuevo Leon, and Zacatecas, characterized by arid climates, were found unsuitable for *A. fraterculus* in all our predictions. The presence of *A. fraterculus* in the United States was previously reported in southern Texas, where the species appears to be restricted [89]. However, our future projections suggest that this pest can spread to new suitable habitats. Therefore, despite the low suitability of the new regions for the establishment of *A. fraterculus*, it remains crucial to monitor the potential expansion of this pest in the U.S.

Overall, our results predict an expansion in the suitable areas for *A. fraterculus* in temperate regions and some tropical areas across the Americas in the coming years. These predictions support our first hypothesis about the pest's potential distribution. The models suggest this expansion could occur despite some regions becoming less suitable due to climate change. However, additional studies will be needed to determine if the predicted suitable areas for this pest overlap with compatible areas for the cultivation of susceptible crops. Our continental-scale projections provide valuable insights into the pest distribution, but we recognize that local microclimates, agricultural practices, and the presence of different pest morphotypes are essential to determine the establishment of this pest. Nevertheless, monitoring efforts should prioritize regions identified as suitable in our models, especially if valuable host crops are cultivated in those areas.

### The fruit fly parasitoid, *Doryctobracon areolatus* as a promising control agent in temperate regions

Suitable areas for *D. areolatus* are predicted in arid and temperate regions, while tropical climates and high-altitude regions seem to be unsuitable or low-suitable for the establishment of this species. Similar results were reported by Núñez-Campero et al. [90], who built SDM for *D. areolatus* in the current period and found that this species is associated with semiarid conditions while high humidity decreases suitability for its establishment. In their projections, precipitation of the driest quarter (BIO17) and precipitation of the coldest quarter (BIO19) were the variables with the greatest contribution to the model, with values of 18% and 33%, respectively. Nonetheless, in our models, the bioclimatic variables associated with temperature and precipitation had the highest contribution values; this discrepancy could be related to different methodologies to construct the models and to different geographic areas of study. However, previous studies demonstrate that the best thermal conditions for the growth of *D. areolatus* are between 22 and 25 °C [91]. Also, studies conducted at different elevations have demonstrated that the parasitism rates of *D. areolatus* are significantly reduced at high altitudes, supporting our results.

In Brazil, certain areas classified by Beck et al. [78] as tropical rainforests in the northern regions of the country were predicted as low suitable or unsuitable for *D. areolatus*, particularly in the Amazonas state (Fig 3). The limited suitability of this parasitoid species in tropical rainforest regions can be attributed to the high precipitation levels in these areas, which affect the population dynamics and reduce the parasitism capacity [5,92]. Our current and future projections indicate that the temperate zones in southern and southeastern Brazil are more suitable for the establishment of *D. areolatus*. These regions are also suitable for the establishment of *A. fraterculus,* suggesting that *D. areolatus* could be a good biological control agent for this pest (S1 Fig). For example, in the state of Rio Grande do Sul, the parasitoid was found in association with *A. fraterculus*, exhibiting parasitism rates between 8% and 25% [93,94].

Moreover, overlapping suitable areas for *A. fraterculus* and *D. areolatus* were predicted in other countries across Latin America (Fig 4; S1 Fig). For instance, in Peru, Colombia, Venezuela, Ecuador, and Bolivia, the parasitoid *D. areolatus* has been found in association with *A. fraterculus* [20,95]. In Peru, some suitable areas for *D. areolatus* were predicted in the "la Sierra" region, which is heavily planted with fruit crops and is characterized in part by a temperate climate [78,96]. Additionally, a study conducted by Salazar-Mendoza et al. [97] in Oxapampa, Pasco (low elevation in la Sierra region), found 11.76% of *A. fraterculus* parasitized by *D. areolatus* in loquat (*Eriobotrya japonica* Lindl.). However, our predictions for future climate scenarios show low suitability of these regions for the establishment of the parasitoid *D. areolatus*. Therefore, we suggest the use of other biocontrol agents to manage *A. fraterculus* in these regions in the future.

Moderately and highly suitable areas in current and future projections were predicted in the valley of the Andes Colombia, specifically in the Departments of Huila, Valle del Cauca, and Tolima (Fig 3). The presence of *D. areolatus* in the Department of Valle del Cauca was previously reported by Carrejo and González [98] preying on *Anastrepha obliqua* (Macquart, 1835) in mango crops. Also, *D. areolatus* in Colombia seems to be more common in areas of low elevations [99], which coincides with our predictions of moderate to low suitability in the Atlantic Coast and Plains regions of Colombia. Therefore, the parasitoid *D. areolatus* could help mitigate the effect of *A. fraterculus* and other *Anastrepha* species in crops planted in low-altitude areas.

In Mexico, we found suitable areas for *D. areolatus* in regions characterized by a low elevation and a tropical savannah climate [78]. Previous studies reported that *D. areolatus* was the most predominant parasitoid across various host species of *Anastrepha* in the states of Vera Cruz and Chiapas [100]. Also, Sivinski et al. [101] found that *D. areolatus* was more frequent in low altitudes in the state of Veracruz, which supports our findings.

In the United States of America, the parasitoid *D. areolatus* was introduced in 1969 in Florida to control the Caribbean fruit fly, *Anastrepha suspensa* (Loew, 1862) [102]. This parasitoid was successfully established in Florida except in the southern part of the state. Eitam et al. [103] pointed out that the absence of *D. areolatus* in this region can be attributed to competitive displacement due to the introduction of *D. longicaudata*. These authors also found that *D. areolatus* was more common in sites with lower mean temperatures, where greater seasonality and less homogeneous host distribution favor this species.

Collectively, our results suggest that the native parasitoid *D. areolatus* is a promising biological control agent against *Anastrepha* fruit flies, particularly in temperate and arid regions, where its establishment and parasitism rates are highest. Tropical and high-altitude areas seem to be unsuitable or low-suitable due to excessive humidity and cooler temperatures, a trend reinforced by future climate projections. The suitable areas for the establishment of *D. areolatus* are predicted to increase between 4.72 and 8.13% in future climate change scenarios, contradicting our second hypothesis of no expansion in suitable areas for this species. This suggests that climate change could expand suitable areas also for pests' parasitoids, reinforcing the viability of biological control programs in the future.

### The fruit fly parasitoid, *Diachasmimorpha longicaudata* as a promising control agent in tropical regions

According to our projections, highly and moderately suitable areas for *D. longicaudata* have tropical climates, including areas characterized as tropical rainforests like the Amazon region. Low suitability or unsuitability is observed in areas with temperate or arid climate, such as the South of Brazil, Paraguay, along the Andes mountains, and certain states in Northwest and Northeast Mexico (Fig 5). Previous studies demonstrated that the optimum temperature for the development of immature stages and fecundity of *D. longicaudata* is between 25 ℃ and 27 ℃, with complete development at 30 ℃ in the hosts *Bactrocera dorsalis* (Hendel, 1912) and *C. capitata* [104–106]. In another study developed by Meirelles et al. [104], regions with an average temperature higher than 25 ℃ were more favorable for establishing *D. longicaudata* as a control agent in the field. Also, according to Harbi et al. [105], temperatures between 16 ℃ and 24 ℃ and relative humidity between 45% and 60% are ideal for parasitism by *D. longicaudata*. According to the same authors, a decrease in relative humidity reduces parasitism rates, while an increase in temperature increases parasitism. These findings collectively suggest that tropical regions provide more suitable conditions for the successful establishment and performance of *D. longicaudata*.

In Northern Brazil, areas suitable for *D. longicaudata* overlap with those of the pest *A. fraterculus* (Fig 6; S2 Fig). However, this tropical region has low suitability for the pest, suggesting that *D. longicaudata* could be a good biological control agent of other tephritids, such as *Bactrocera carambolae* (Drew & Hancock, 1994) (Diptera: Tephritidae) in these zones [107]. Moderately suitable areas were also identified in Southeast Brazil, particularly in São Paulo, where *D. longicaudata* could aid in tephritid control in citrus orchards. Paranhos et al. [33] reported that summer releases enhance parasitism, while winter conditions allow high survival rates despite heavy rainfall. In contrast, a low adaptation of *D. longicaudata* was observed before in the state of Amapá in North Brazil [108], which aligns with our current projections for some areas classified as low suitable. However, future predictions indicate that Southeast and North Brazil may become moderately to highly suitable for the establishment of this parasitoid, suggesting that *D. longicaudata* could be used in forthcoming biological control programs of tephritid fruit flies. In the south of Brazil, previous studies developed in Rio Grande do Sul, demonstrated that the parasitoid has limited tolerance to winter, leading to less number of generations [104]. Therefore, other biocontrol agents that are better adapted to the environmental conditions of southern Brazil, such as *D. areolatus,* may be better options for controlling *A. fraterculus*.

In Peru, unlike *D. areolatus*, high and moderately suitable areas were projected across the entire region of "la Selva" and low suitable areas in almost the entire region of "la Sierra" (Fig 4 and 6). This suggests that the parasitoid *D. longicaudata* could be an important biocontrol agent for *A. fraterculus* in Peru's fruit-growing regions. Similarly, in Colombia, *D. longicaudata* may be an effective biological control agent of *A. fraterculus* particularly in tropical regions that are predicted to be unsuitable or low suitable for the establishment of *D. areolatus.*

In future scenarios, suitable areas for the establishment of *D. longicaudata* were predicted across all Mesoamerica (Fig 5). The coastal regions of Mexico, characterized by a tropical climate, were predicted as suitable for the establishment of *D. longicaudata*, while the temperate and arid regions of Northeast, and Central Mexico were unsuitable. Our predictions indicate an expansion of suitable areas for *D. longicaudata* in future climate change scenarios, which can be attributed to a projected shift in certain Mexican regions from temperate to tropical [78]. This change could benefit the expansion of *D.*

*longicaudata* populations in the region. Some suitable areas for *D. longicaudata* in Mexico, such as the southern part of Chiapas cultivated with mango, exhibit parasitism rates between 0.44% and 29.23% on other *Anastrepha* species [109]. Augmentative biological control of *Anastrepha* species in this region has shown that *D. longicaudata* can reduce pest populations without negatively affecting parasitoid local richness [110,111]. Therefore, *D. longicaudata* is a promising biocontrol agent for *A. fraterculus* and other *Anastrepha* species in the tropical climate areas in Mexico.

In the United States, *D. longicaudata* was introduced in 1988 to control *C. capitata* in Florida. This parasitoid reduced the fruit fly populations and is now established in the region [103,112]. The distribution of *D. longicaudata* proposed a decade ago aligns closely with the suitable areas predicted by our model [103]. According to our results, both *D. areolatus* and *D. longicaudata* could be used for the control of *A. fraterculus* in the State of Florida (Fig 4 and 6). However, if the pest spreads to other northern states in the future, where conditions are unsuitable for both parasitoids according to our predictions, the use of alternative control tactics or biocontrol agents may become necessary.

In summary, suitable habitats for *D. longicaudata* may expand in tropical regions under future climate scenarios, contradicting our second and third hypotheses. This suggests that non-native parasitoids like *D. longicaudata* could occupy a broader geographical range than native species under climate change. Importantly, *D. longicaudata* could serve as an effective biocontrol agent against fruit flies, particularly in areas where *D. areolatus* is unsuitable, thereby nearly covering the entire potential range of the pest.

## Conclusions

Our study highlights the potential expansion of the South American fruit fly *A. fraterculus* in the future under climate change, with a potential increase in fruit damage in countries like Brazil and Colombia. Suitable areas for the establishment and dispersion of this pest in the Americas were mostly associated with temperate and some tropical climate regions. Our findings also suggest an increase of suitable areas for the establishment of the parasitoids *D. areolatus* and *D. longicaudata*. The exotic parasitoid *D. longicaudata* could have a greater geographical expansion than *D. areolatus* under future climate change scenarios. Suitable areas for the establishment of *D. longicaudata* were more common in tropical climate regions, especially in tropical rainforests, whereas suitable areas for the establishment of *D. areolatus* were more common in temperate climates. Due to the ability of these parasitoids to thrive in different climates, they could be used as biocontrol agents for *A. fraterculus* in almost all suitable areas for the establishment of this pest. Future studies using SDMs for each *A. fraterculus* morphotype can be informative for designing control tactics at a local level in the Americas. The use of SDMs to predict the establishment of biocontrol agents and pests under climate change is informative and useful for designing pest management strategies.

## Supporting information

**S1 Fig. Overlapping suitable predicted areas for *A. fraterculus* and *D. areolatus* in the future.** Overlapping suitable areas were predicted for *A. fraterculus* and *D. areolatus* under two future climatic scenarios (SSP2–4.5 and SSP5–8.5) and two time periods (2021–2040 and 2041–2060).
(TIF)

**S2 Fig. Overlapping suitable predicted areas for *A. fraterculus* and *D. longicaudata* in the future.** Overlapping suitable areas were predicted for *A. fraterculus* and *D. longicaudata* under two future climatic scenarios (SSP2–4.5 and SSP5–8.5) and two time periods (2021–2040 and 2041–2060).
(TIF)

**S1 Table. Global Climate Models (GCM) from the Coupled Model Intercomparison Project Phase 6 (CMIP6) used in our Random Forest model construction.**
(XLSX)

**S2 Table. Extension of suitable and unsuitable areas predicted for the pest, *A. fraterculus* and the parasitoids *D. areolatus* and *D. longicaudata* across the Americas.**
(XLSX)

## Acknowledgments

We thank the two anonymous reviewers who helped with the improvement of the manuscript.

## Author contributions

**Conceptualization:** Jesús H. Gómez-Llano, Fábio L. Galvão-Silva, Marco Silva Gottschalk, Dori Edson Nava.

**Data curation:** Jesús H. Gómez-Llano.

**Formal analysis:** Jesús H. Gómez-Llano.

**Funding acquisition:** Dori Edson Nava.

**Investigation:** Jesús H. Gómez-Llano, Fábio L. Galvão-Silva, Flor E. Acevedo, Dori Edson Nava.

**Methodology:** Jesús H. Gómez-Llano.

**Software:** Jesús H. Gómez-Llano, Fabio Castro-Llanos.

**Supervision:** Marco Silva Gottschalk.

**Validation:** Jesús H. Gómez-Llano, Fábio L. Galvão-Silva, Flor E. Acevedo.

**Visualization:** Jesús H. Gómez-Llano, Fábio L. Galvão-Silva, Flor E. Acevedo, Fabio Castro-Llanos.

**Writing – original draft:** Jesús H. Gómez-Llano, Fábio L. Galvão-Silva, Flor E. Acevedo.

**Writing – review & editing:** Jesús H. Gómez-Llano, Flor E. Acevedo, Dori Edson Nava.

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
