## [Decision Letter · Decision Letter 0]

PONE-D-24-44826Biological Control under Climate Change: Distribution Patterns of the South American Fruit Fly, Anastrepha fraterculus and Two of its Parasitoids in the AmericasPLOS ONE

Dear Dr. Gomez Llano,

Thank you for submitting your manuscript to PLOS ONE. After careful consideration, we feel that it has merit but does not fully meet PLOS ONE’s publication criteria as it currently stands. Therefore, we invite you to submit a revised version of the manuscript that addresses the points raised during the review process.

The paper contains useful information following an interesting approach. However, evaluating the ms myself I m rather aligned with Reviewer #1. Without pointing into details at the current stages, I found the Results and Discussion section difficult to follow, wordy and with details that can not be followed. Also, data presentation should be reconsidered and become more comprehensive. Apparently, this needs to be thoroughly revised. Also, I would suggest to separate Results from discussion. Also, to improve the ms comments of both reviewers, especially those of Reviewer 1, should be carefully addressed. The list o References cited should be also throughly edited to align with the style of Plos One.  

We look forward to receiving your revised manuscript.

Kind regards,

Nikos T. Papadopoulos

Academic Editor

PLOS ONE

Journal Requirements:

3. Thank you for stating the following in the Acknowledgments Section of your manuscript: [We thank the anonymous reviewers who helped with the improvement of the manuscript, and the Coordination for the Improvement of Higher Education Personnel – Brazil (CAPES) by financial support. The author D.E. Nava is thankful for the financial support of Conselho Nacional de Desenvolvimento Científico e Tecnológico – CNPq, #310233/2020-8] We note that you have provided funding information that is not currently declared in your Funding Statement. However, funding information should not appear in the Acknowledgments section or other areas of your manuscript. We will only publish funding information present in the Funding Statement section of the online submission form. Please remove any funding-related text from the manuscript and let us know how you would like to update your Funding Statement. Currently, your Funding Statement reads as follows: [The authors received no specific funding for this work.] Please include your amended statements within your cover letter; we will change the online submission form on your behalf.

4. We note that [Figures 2-4] in your submission contain [map/satellite] images which may be copyrighted. All PLOS content is published under the Creative Commons Attribution License (CC BY 4.0), which means that the manuscript, images, and Supporting Information files will be freely available online, and any third party is permitted to access, download, copy, distribute, and use these materials in any way, even commercially, with proper attribution. For these reasons, we cannot publish previously copyrighted maps or satellite images created using proprietary data, such as Google software (Google Maps, Street View, and Earth). For more information, see our copyright guidelines: http://journals.plos.org/plosone/s/licenses-and-copyright.

a. You may seek permission from the original copyright holder of Figures 2-4 to publish the content specifically under the CC BY 4.0 license.   

5. Please include captions for your Supporting Information files at the end of your manuscript, and update any in-text citations to match accordingly. Please see our Supporting Information guidelines for more information: http://journals.plos.org/plosone/s/supporting-information .

Additional Editor Comments:

The paper contains useful information following an interesting approach. However, evaluating the ms myself I m rather aligned with Reviewer #1. Without pointing into details at the current stages, I found the Results and Discussion section difficult to follow, wordy and with details that can not be followed. Also, data presentation should be reconsidered and become more comprehensive. Apparently, this needs to be thoroughly revised. Also, I would suggest to separate Results from discussion. Also, to improve the ms comments of both reviewers, especially those of Reviewer 1, should be carefully addressed. The list o References cited should be also throughly edited to align with the style of Plos One.

Reviewers' comments:

Reviewer's Responses to Questions

**Comments to the Author**

1. Is the manuscript technically sound, and do the data support the conclusions?

Reviewer #1: Yes

Reviewer #2: Partly

2. Has the statistical analysis been performed appropriately and rigorously? 

Reviewer #1: Yes

Reviewer #2: N/A

3. Have the authors made all data underlying the findings in their manuscript fully available?

Reviewer #1: Yes

Reviewer #2: No

4. Is the manuscript presented in an intelligible fashion and written in standard English?

Reviewer #1: Yes

Reviewer #2: No

5. Review Comments to the Author

Reviewer #1: The reviewer found the manuscript to be timely and of considerable general interest. Here are some minor comments.

References 1-5 are outdated and newer articles with more information about recent climate change are readily available and free. It would be preferable to keep these but also add 5 newer articles to flesh out this part of the introduction.

Line 88 “agent against such species”

Lines 261-266. The reviewer had difficulty sorting out what was meant by relative contribution because the number of individuals comprising each individual code is not well specified. Perhaps information could be added to Table 1 about this.

Line 288. Should this be Fig. or Figure? At first, the reviewer could not find Figure 2.

Line 365 Reference [88]. Steck et al 1980, Proc Entomol Soc. Washington, is a more recent reference to presence in the US.

Reviewer #2: General comments:

The manuscript is not written well and need intensive English editing (language rules, syntax, preposition words). In addition, there are many redundant arguments within text which make reading awkward.

Due to the general scope of the manuscript, its better to present data with tables and graphs. The results section contains mass textual descriptions of future regional distributions within Brazil, which are not common for the global reader and cannot be accessible in the presented maps. It's suggested to use geographical and climatic characteristics (i.e., arid, temperate, high altitude etc') instead.

The authors describe in details the expected distribution changes within countries. The general phenomena are increase in all three species distribution which allow to keep biological control activities. The authors however do not elaborate the potential biological responses due to the significant differences between the two parasitoied species (general increase of 10-13% with A. fraterculus, 4-8% with D. longicaudata and 22-38% with D. arealotus upon the different model scenarios). As authors cite, there are already evident from the field on the effects of climate change on the focal species. If the model may contribute to the potential activity of A. fraterculus parasitoids, its expected to discusses it thorough.

Examples:

Introduction:

Row 49: its better to delete " and their natural enemies" , since you use (row 51) "Similarly" and describe the effect on natural enemies.

Row 52: " than on insect pests themselves" its not clear why you repeat on what is written before (Row 49)

Row 63: " Currently, many SDMs are being developed using correlative models" repeat previous argument (row 60).

Row 64: unlike the "SDMs" MaxEnt and Bioclim, "Random forest" is a machine learning algorithm and not a species distribution Model per se.

Row 68: the author claims that " Currently, the focus of SDMs is not only to understand the effects of climate change on agricultural pests…." Need to be justified by the developers of MaxEnt and Bioclim which are not been credited within this manuscript.

Methods:

The author used random pseudo-absences locations where it is assumed that there is no record or presence of the species, thus providing a point of comparison with the presence records of each species with a ratio of 2:1 pseudo-absence to presence locations generation. Its important to understand 1. what is the spatial scale from which those pseudo locations were chosen (may affect model outcomes)? 2. What is the rational for choosing ratio of 2:1 pseudo-absence to presence locations?

Results

Using previous findings (Selivon et al) of different distribution of three Brazilian morphotypes, the authors argue that " our results suggest that the expansion of suitable areas for A. fraterculus could be associated with specific adaptations of morphotypes.". since the model is generalized for the 10 mentioned A. fraterculus morphotypes, this link needs at least supporting evident.

6. PLOS authors have the option to publish the peer review history of their article (what does this mean? ). If published, this will include your full peer review and any attached files.

**Do you want your identity to be public for this peer review?** For information about this choice, including consent withdrawal, please see our Privacy Policy .

Reviewer #1: No

Reviewer #2: No

---

## [Author Response · Author response to Decision Letter 1]

8 Jan 2025

I would like to thank the editor and reviewers for their valuable contributions, which have been taken into account. Broadly speaking, the suggested changes in the manuscript were implemented; the Results and Discussion sections were separated, the English writing was improved, and figures were added to facilitate the understanding of the results. Responses to each suggestion can be found in the document 'Response to Reviewers'. However, I will add the responses here.

Edittor comments:

The paper contains useful information following an interesting approach. However, evaluating the ms myself I m rather aligned with Reviewer #1. Without pointing into details at the current stages, I found the Results and Discussion section difficult to follow, wordy and with details that can not be followed. Also, data presentation should be reconsidered and become more comprehensive. Apparently, this needs to be thoroughly revised. Also, I would suggest to separate Results from discussion. Also, to improve the ms comments of both reviewers, especially those of Reviewer 1, should be carefully addressed. The list o References cited should be also throughly edited to align with the style of Plos One.

Reply:

We appreciate the observations, we separate the Results and Discussion sections to facilite the compression, also we considerate to include two new figures to be more comprehensive, and the references were edited to aling with Plos one style

Journal Requirements:

3. Thank you for stating the following in the Acknowledgments Section of your manuscript: [We thank the anonymous reviewers who helped with the improvement of the manuscript, and the Coordination for the Improvement of Higher Education Personnel – Brazil (CAPES) by financial support. The author D.E. Nava is thankful for the financial support of Conselho Nacional de Desenvolvimento Científico e Tecnológico – CNPq, #310233/2020-8] We note that you have provided funding information that is not currently declared in your Funding Statement. However, funding information should not appear in the Acknowledgments section or other areas of your manuscript. We will only publish funding information present in the Funding Statement section of the online submission form. Please remove any funding-related text from the manuscript and let us know how you would like to update your Funding Statement. Currently, your Funding Statement reads as follows: [The authors received no specific funding for this work.] Please include your amended statements within your cover letter; we will change the online submission form on your behalf.

4. We note that [Figures 2-4] in your submission contain [map/satellite] images which may be copyrighted. All PLOS content is published under the Creative Commons Attribution License (CC BY 4.0), which means that the manuscript, images, and Supporting Information files will be freely available online, and any third party is permitted to access, download, copy, distribute, and use these materials in any way, even commercially, with proper attribution. For these reasons, we cannot publish previously copyrighted maps or satellite images created using proprietary data, such as Google software (Google Maps, Street View, and Earth). For more information, see our copyright guidelines: http://journals.plos.org/plosone/s/licenses-and-copyright.

Reply:

1. We have updated the style of the paper to comply with the PLOS ONE formatting guidelines.

2. Changes have been made to sections 2 and 3 of the manuscript. Additionally, we have resubmitted the 'Funding Information' and 'Financial Disclosure' sections as requested.

3. The maps were created using RStudio with the rnaturalearth package. The methodology has been revised to clarify this process.

4. The supporting information captions were include at the end of the manuscript.

Review Comments to the Author

Reviewer #1: The reviewer found the manuscript to be timely and of considerable general interest. Here are some minor comments.

References 1-5 are outdated and newer articles with more information about recent climate change are readily available and free. It would be preferable to keep these but also add 5 newer articles to flesh out this part of the introduction.

Reply: Changes in references were made

Line 88 “agent against such species”

Reply: We have corrected the English throughout the manuscript.

Lines 261-266. The reviewer had difficulty sorting out what was meant by relative contribution because the number of individuals comprising each individual code is not well specified. Perhaps information could be added to Table 1 about this.

Reply: The Relative Variable Importance ranks predictors based on their impact on model performance, showing how each predictor contributes to improvements when splits are made across the entire forest

Line 288. Should this be Fig. or Figure? At first, the reviewer could not find Figure 2.

Reply: Changes were made

Line 365 Reference [88]. Steck et al 1980, Proc Entomol Soc. Washington, is a more recent reference to presence in the US.

Reply: No recent references have been found regarding the presence of the species in the United States.

Reviewer #2: General comments:

The manuscript is not written well and need intensive English editing (language rules, syntax, preposition words). In addition, there are many redundant arguments within text which make reading awkward.

Due to the general scope of the manuscript, its better to present data with tables and graphs. The results section contains mass textual descriptions of future regional distributions within Brazil, which are not common for the global reader and cannot be accessible in the presented maps. It's suggested to use geographical and climatic characteristics (i.e., arid, temperate, high altitude etc') instead.

The authors describe in details the expected distribution changes within countries. The general phenomena are increase in all three species distribution which allow to keep biological control activities. The authors however do not elaborate the potential biological responses due to the significant differences between the two parasitoied species (general increase of 10-13% with A. fraterculus, 4-8% with D. longicaudata and 22-38% with D. arealotus upon the different model scenarios). As authors cite, there are already evident from the field on the effects of climate change on the focal species. If the model may contribute to the potential activity of A. fraterculus parasitoids, its expected to discusses it thorough.

Reply: The manuscript has undergone English editing, and the Results and Discussion sections have been separated. The climatic characteristics used in the manuscript are accessible. The inclusion of the Basil regions was necessary because, in our study, Brasil was the country with the largest suitable areas for the pest. Regarding the increase in suitable areas mentioned by the reviewer, we believe the extension of suitable areas is more important for predicting locations where biological control programs can be implemented.

Examples:

Introduction:

Row 49: its better to delete " and their natural enemies" , since you use (row 51) "Similarly" and describe the effect on natural enemies.

Row 52: " than on insect pests themselves" its not clear why you repeat on what is written before (Row 49)

Row 63: " Currently, many SDMs are being developed using correlative models" repeat previous argument (row 60).

Row 64: unlike the "SDMs" MaxEnt and Bioclim, "Random forest" is a machine learning algorithm and not a species distribution Model per se.

Row 68: the author claims that " Currently, the focus of SDMs is not only to understand the effects of climate change on agricultural pests…." Need to be justified by the developers of MaxEnt and Bioclim which are not been credited within this manuscript.

Reply: We have corrected the English throughout the manuscript.

Methods:

The author used random pseudo-absences locations where it is assumed that there is no record or presence of the species, thus providing a point of comparison with the presence records of each species with a ratio of 2:1 pseudo-absence to presence locations generation. Its important to understand 1. what is the spatial scale from which those pseudo locations were chosen (may affect model outcomes)? 2. What is the rational for choosing ratio of 2:1 pseudo-absence to presence locations?

Reply:

1. "The scale of the pseudo-absences was set at 2.5 arc-minutes.

2. Typically, the minimum number of pseudo-absences used in random forest species distribution models (SDM) is at least equal to the number of occurrences. It is also recommended to use 10 replications when there are fewer than 10,000 occurrences. To improve the accuracy of our predictions, we increased the number of pseudo-absences and conducted at least 10 replications. The AUC values of our models indicate that our predictions are accurate.

Results

Using previous findings (Selivon et al) of different distribution of three Brazilian morphotypes, the authors argue that " our results suggest that the expansion of suitable areas for A. fraterculus could be associated with specific adaptations of morphotypes.". since the model is generalized for the 10 mentioned A. fraterculus morphotypes, this link needs at least supporting evident.

Reply:

This misunderstanding arose during the writing process. We have corrected the English to better reflect our results. What we intended to convey is that our study evaluates the species A. fraterculus without the different morphotypes.

---

## [Decision Letter · Decision Letter 1]

PONE-D-24-44826R1Biological Control under Climate Change: Distribution Patterns of the South American Fruit Fly, Anastrepha fraterculus and Two of its Parasitoids in the AmericasPLOS ONE

Dear Dr. Acevedo,

Thank you for submitting your manuscript to PLOS ONE. After careful consideration, we feel that it has merit but does not fully meet PLOS ONE’s publication criteria as it currently stands. Therefore, we invite you to submit a revised version of the manuscript that addresses the points raised during the review process.

 This is a much improved version, however, the ms still needs additional improvements. Please careful edit your ms to remove grammatical error and improve other parts (e.g. quotation of references cited). Comments of both reviewers should be carefully considered and addressed. Authors may also consider to further elaborate either on maps or in the text on the expected fruit crops that would come under attack due to climate change. 

We look forward to receiving your revised manuscript.

Kind regards,

Nikos T. Papadopoulos

Academic Editor

PLOS ONE

Journal Requirements:

Reviewers' comments:

Reviewer's Responses to Questions

**Comments to the Author**

1. If the authors have adequately addressed your comments raised in a previous round of review and you feel that this manuscript is now acceptable for publication, you may indicate that here to bypass the “Comments to the Author” section, enter your conflict of interest statement in the “Confidential to Editor” section, and submit your "Accept" recommendation.

Reviewer #1: (No Response)

Reviewer #3: All comments have been addressed

2. Is the manuscript technically sound, and do the data support the conclusions?

Reviewer #1: Yes

Reviewer #3: Yes

3. Has the statistical analysis been performed appropriately and rigorously? 

Reviewer #1: Yes

Reviewer #3: Yes

4. Have the authors made all data underlying the findings in their manuscript fully available?

Reviewer #1: Yes

Reviewer #3: Yes

5. Is the manuscript presented in an intelligible fashion and written in standard English?

Reviewer #1: Yes

Reviewer #3: Yes

6. Review Comments to the Author

Reviewer #1: The manuscript has been improved but grammatical errors remain at the following lines and should be corrected before publication.

Line 50- should be “Therefore, increases in mean temperature, fewer occurrences of freezing temperatures, “

Line 68- should be “This approach may help entomologists to predict”.

Line 79- “MASL (meters above sea level)”.

Line 110 delete “in the future”

Line 398 delete “While” or use a comma instead of a period.

Line 41 possibly should be “in the U.S. and Mexico”

Line 533 delete While” as in line 398.

Line 581 “Florida. The parasitoid reduced”

Reviewer #3: The manuscript entitled “Biological control under climate change: Distribution patterns of the South American fruit fly, Anastrepha fraterculus and two of its parasitoids in the Americas” uses species distribution models (SDM) to predict the potential geographic range of a pest, its biological control agents, and the areas that overlap between the two.

The manuscript is clear and acceptable for publication. Here are some minor suggestions or comments:

In Introduction: The three hypotheses (L113 – 119) are interesting but are never discussed further.

Materials and methods: (L123-130) How many occurrences per species were found and used to build the SDM analysis? Are you sure you can find occurrences of all three species in the Vertebrate Network? Have you sorted or selected the occurrences (according to their reliability, for example)?

Results:

In general: Occupation zones are described by countries, regions (Amazon region, Mesoamerican countries, Orinoquia region....) and geological formations (Andes mountains). These names are never indicated on any map, which makes them difficult for non-specialist readers to understand.

L247-251: “the largest extension of suitable areas”. This paragraph requires clarification. You mention "largest extension of suitable areas", but it is unclear whether the values refer to current or future occupancy or an extension factor. From Table 2, it looks like occupancy between 2041 and 2060 according to the SS95-8.5 scenario, although not all the values are identical.

Fig 1 legend: L291: Remove “Extension of”.

Fig 2 legend: L297: Change “(B-D)” by (B-E). Idem for fig 3 and fig 5

Discussion: The three main parts of the discussion contain a substantial amount of information. It may be advisable to conclude each section with a summary of the key points (a take-home message).

Bibliography: In some refs, each word is capitalized. Species names are not always italicized.

7. PLOS authors have the option to publish the peer review history of their article (what does this mean? ). If published, this will include your full peer review and any attached files.

**Do you want your identity to be public for this peer review?** For information about this choice, including consent withdrawal, please see our Privacy Policy .

Reviewer #1: No

Reviewer #3: No

---

## [Author Response · Author response to Decision Letter 2]

29 Apr 2025

Response to reviewers

Editor comments:

Thank you for submitting your manuscript to PLOS ONE. After careful consideration, we feel that it has merit but does not fully meet PLOS ONE’s publication criteria as it currently stands. Therefore, we invite you to submit a revised version of the manuscript that addresses the points raised during the review process.

This is a much improved version, however, the ms still needs additional improvements. Please careful edit your ms to remove grammatical error and improve other parts (e.g. quotation of references cited). Comments of both reviewers should be carefully considered and addressed. Authors may also consider to further elaborate either on maps or in the text on the expected fruit crops that would come under attack due to climate change.

Reply: We appreciate the editor’s observations. The grammatical errors in the manuscript have been corrected, and the references have been revised to comply with the journal’s requirements. Regarding the fruit crops potentially affected by the expansion of A. fraterculus, we mentioned some of these crops in the manuscript. However, we acknowledge that this pest has over 160 host species. To address this, we have added a paragraph at the end of the first section of the discussion, emphasizing the importance of focusing research on crops in high-risk regions.

Review Comments to the Author:

1. Reviewer #1: General comments:

The manuscript has been improved but grammatical errors remain at the following lines and should be corrected before publication.

Line 50- should be “Therefore, increases in mean temperature, fewer occurrences of freezing temperatures”

Response: The sentence was corrected (line 52)

Line 68- should be “This approach may help entomologists to predict”.

Response: The sentence was corrected (line 71)

Line 79- “MASL (meters above sea level)”.

Response: The sentence was corrected (line 83)

Line 110 delete “in the future”

Response: The sentence was corrected (line 123)

Line 398 delete “While” or use a comma instead of a period.

Response: The sentence was corrected (line 446)

Line 41 possibly should be “in the U.S. and Mexico”

Response: Maybe the Line wasn’t written right by the reviewer, because do not match the line 41 with the suggestion.

Line 533 delete While” as in line 398.

Response: The sentence was corrected (line 623)

Line 581 “Florida. The parasitoid reduced”

Response: The sentence was corrected (line 690)

We sincerely appreciate the time and careful attention put into reviewing our manuscript. We have implemented all the grammatical corrections you identified,

2. Reviewer #3: General comments:

The manuscript entitled “Biological control under climate change: Distribution patterns of the South American fruit fly, Anastrepha fraterculus and two of its parasitoids in the Americas” uses species distribution models (SDM) to predict the potential geographic range of a pest, its biological control agents, and the areas that overlap between the two.

The manuscript is clear and acceptable for publication. Here are some minor suggestions or comments:

Introduction comments: The three hypotheses (L113 – 119) are interesting but are never discussed further.

Reply: In response to the reviewer's comment, we have added dedicated discussion paragraphs after each relevant section to explicitly address and evaluate the three hypotheses. See lines 525-534, 609-617, and 704-709.

Materials and methods comments: (L123-130) How many occurrences per species were found and used to build the SDM analysis? Are you sure you can find occurrences of all three species in the Vertebrate Network? Have you sorted or selected the occurrences (according to their reliability, for example)?

Reply: The number of occurrences used in the model’s construction has been added to the methods section (lines 143-148). Regarding the vertebrate network database, no records were initially available for the species in this database. To obtain these records, we used the “occ” function from the “spocc” package in RStudio, following the code provided by its authors, and included all recommended databases. The occurrences were then filtered to remove duplicates. Due to the limited number of available records for the parasitoids, we retained all occurrences in the final model construction.

Results comments: In general: Occupation zones are described by countries, regions (Amazon region, Mesoamerican countries, Orinoquia region....) and geological formations (Andes mountains). These names are never indicated on any map, which makes them difficult for non-specialist readers to understand.

Reply: We appreciate the suggestion. While we did not label specific zones (e.g., countries, regions like the Amazon or Orinoquía, or geological formations like the Andes) on the maps, we intentionally described them using cardinal directions (e.g., northern, southern) within each country or area. This approach was taken to emphasize the significance of these regions in relation to pest expansion and to guide research focus accordingly. However, we acknowledge that adding geographical labels could improve clarity for non-specialist readers, and we will consider this for future work.

Other comments: L247-251: “the largest extension of suitable areas”. This paragraph requires clarification. You mention "largest extension of suitable areas", but it is unclear whether the values refer to current or future occupancy or an extension factor. From Table 2, it looks like occupancy between 2041 and 2060 according to the SS95-8.5 scenario, although not all the values are identical.

Reply: The paragraph has been modified to specify that the values refer to projected occupancy under the SSP5-8.5 scenario (2041-2060). See lines 273-277

Fig 1 legend: L291: Remove “Extension of”.

Reply: "Extension of" has been removed from the Fig 1 legend (Line 330)

Fig 2 legend: L297: Change “(B-D)” by (B-E). Idem for fig 3 and fig 5.

The figure references have been corrected to (B-E) in Figs 2, 3, and 5 legends. (lines 336, 370, 429).

Discussion comments: The three main parts of the discussion contain a substantial amount of information. It may be advisable to conclude each section with a summary of the key points (a take-home message).

Reply: We appreciate the reviewer's suggestion. In response, we have added concise summary statements at the end of each discussion section to highlight the key findings and main conclusions of our study.

Bibliography comments: In some refs, each word is capitalized. Species names are not always italicized.

Reply: Regarding our bibliography formatting, we have systematically addressed these issues by:

• Standardizing title capitalization to sentence case for all references

• Ensuring proper italicization of all species names throughout the bibliography

These corrections have been implemented across the entire reference section to comply with the journal's style guidelines and maintain proper formatting standards.

We extend our sincere gratitude to the reviewers for their insightful feedback, which has been important in refining this work. We deeply appreciate the careful attention given to our manuscript; each constructive comment has helped strengthen our arguments and enhance clarity. The manuscript has significantly improved through the revision process, and we are truly thankful for the expertise shared throughout the review.

---

## [Editor Report · Decision Letter 2]

Biological control under climate change: Distribution patterns of the South American fruit fly, Anastrepha fraterculus and two of its parasitoids in the Americas

PONE-D-24-44826R2

Dear Dr. Acevedo,

We’re pleased to inform you that your manuscript has been judged scientifically suitable for publication and will be formally accepted for publication once it meets all outstanding technical requirements.

Kind regards,

Nikos T. Papadopoulos

Academic Editor

PLOS ONE

Additional Editor Comments (optional):

there are some typos and minor points that can be easily addressed during proof reading. See some remarks on the attached pdf 

For change in corresponding author and inclusion of Funding please discuss with the editorial office 
---

## [Editor Report · Acceptance letter]

PONE-D-24-44826R2

PLOS ONE

Dear Dr. Acevedo,

I'm pleased to inform you that your manuscript has been deemed suitable for publication in PLOS ONE. Congratulations! Your manuscript is now being handed over to our production team.

Kind regards,

on behalf of

Dr. Nikos T. Papadopoulos

Academic Editor

PLOS ONE